# Performance of Machine Learning Models in Predicting 30-Day General Medicine Readmissions Compared to Traditional Approaches in Australian Hospital Setting

**DOI:** 10.3390/healthcare13111223

**Published:** 2025-05-23

**Authors:** Yogesh Sharma, Campbell Thompson, Arduino A. Mangoni, Rashmi Shahi, Chris Horwood, Richard Woodman

**Affiliations:** 1Department of Acute and General Medicine, Flinders Medical Centre, Adelaide, SA 5042, Australia; chris.horwood@sa.gov.au; 2College of Medicine & Public Health, Flinders University, Adelaide, SA 5042, Australia; arduino.mangoni@flinders.edu.au (A.A.M.); rashmishahi5@hotmail.com (R.S.); richard.woodman@flinders.edu.au (R.W.); 3Discipline of Medicine, University of Adelaide, Adelaide, SA 5005, Australia; campbell.thompson@adelaide.edu.au

**Keywords:** readmissions, prediction models, LACE index, logistic regression, machine learning

## Abstract

**Background/Objectives**: Hospital readmissions are a key quality metric impacting both patient outcomes and healthcare costs. Traditional logistic regression models, including the LACE index (Length of stay, Admission type, Comorbidity index, and recent Emergency department visits), are commonly used for readmission risk stratification, though their accuracy may be limited by non-linear interactions with other clinical variables. This study compared the predictive performance of non-linear machine learning (ML) models with stepwise logistic regression (LR) and the LACE index for predicting 30-day general medicine readmissions. **Methods**: We retrospectively analysed adult general medical admissions at a tertiary hospital in Australia from 1 July 2022 to 30 June 2023. Thirty-two variables were extracted from electronic medical records, including demographics, comorbidities, prior healthcare use, socioeconomic status (SES), laboratory data, and frailty (measured by the Hospital Frailty Risk Score). Predictive models included stepwise LR and four ML algorithms: Least Absolute Shrinkage and Selection Operator (LASSO), random forest, Extreme Gradient Boosting (XGBoost), and artificial neural networks (ANNs). Performance was assessed using the area under the curve (AUC), with comparisons made using DeLong’s test. **Results**: Of 5371 admissions, 1024 (19.1%) resulted in 30-day readmissions. Readmitted patients were older and frailer and had more comorbidities and lower SES. Logistic regression (LR) identified the key predictors of outcomes, including heart failure, alcoholism, nursing home residency, and prior admissions, achieving an AUC of 0.62. LR’s performance was comparable to that of the LACE index (AUC = 0.61) and machine learning models: LASSO (AUC = 0.63), random forest (AUC = 0.60), and artificial neural networks (ANNs) (AUC = 0.60) (*p* > 0.05). However, LR significantly outperformed XGBoost (AUC = 0.55) (*p* < 0.05). **Conclusions**: About one in five general medicine patients are readmitted within 30 days. Traditional LR performed as well as or better than ML models for readmission risk prediction.

## 1. Introduction

Hospital readmissions are widely regarded as a key indicator of the quality and effectiveness of care provided during an initial hospital stay [1]. Globally, health systems use readmission rates not only to compare hospitals’ performance but also as a measure for determining reimbursement, especially in value-based care models [2]. Accurately predicting hospital readmissions offers the potential for targeted interventions that can improve patient outcomes, reduce costs, and enhance the overall quality of care [3].

Recent advances in machine learning (ML) and artificial intelligence (AI) have garnered significant attention for their transformative potential in healthcare [4]. One of the prominent applications of ML is in developing risk stratification models aimed at predicting specific clinical outcomes, such as 30-day hospital readmissions—a critical target for both clinicians and healthcare administrators [5]. Traditional models for predicting readmission risk, including the LACE index [6] and the HOSPITAL score [7], rely on statistical approaches like logistic regression. While these models have been instrumental in clinical decision-making, ML offers the ability to uncover complex, non-linear patterns in large datasets, potentially improving predictive accuracy by accounting for multi-dimensional interactions and intricate data relationships [5,8,9].

The capacity of ML to model complex data structures is particularly promising in the setting of general medical patients, who often present with multiple comorbidities and intricate social determinants of health that are difficult to capture through conventional models [10]. However, the perceived superiority of ML over traditional statistical methods remains a topic of debate [11]. A systematic review of 71 studies found no significant difference in predictive performance between ML models and logistic regression for binary outcomes [11]. Nevertheless, other research [5,12] suggests that in certain contexts, ML may outperform traditional approaches, highlighting the need for further investigation.

The objectives of this study were to evaluate the performance of various ML models in predicting unplanned 30-day hospital general medical readmissions and to compare their accuracy with that of conventional logistic regression and the LACE index. By directly comparing these approaches, this study aimed to provide insight into the relative strengths of ML models in this critical area of healthcare.

## 2. Materials and Methods

This retrospective study included all adult patients (aged ≥ 18 years) admitted under the General Medicine Department at Flinders Medical Centre (FMC) between 1 July 2022 and 30 June 2023. Patients were eligible if they were discharged alive and data were obtained from an electronic medical record (EMR) database. Readmission status was determined based on any unplanned hospital readmission within 30 days of discharge. To account for readmissions occurring at non-index hospitals, readmission data were collected for all metropolitan hospitals in South Australia, as previous research indicates that 15–20% of patients may be readmitted to other hospitals [13]. In cases where patients experienced multiple readmissions, only the first readmission was considered for analysis. The Southern Adelaide Clinical Human Research Ethics Committee (SAC HREC) deemed this study exempt from requiring ethical approval (reference number 4774 dated 14 November 2023).

The primary outcome was all-cause unplanned hospital readmission within 30 days of discharge. Secondary outcomes included the identification of significant predictors of readmission and a comparison of predictive performance between stepwise logistic regression, the LACE index, and machine learning models using the area under the receiver operating characteristic curve (AUC).

### 2.1. Variables Included in This Study

A total of 32 variables were extracted from the EMR database and evaluated in each of the prediction models. These variables are listed in Appendix A. These included age; gender; race; living status; frailty (assessed using the Hospital Frailty Risk Score (HFRS), where a score ≥ 5 classifies a patient as frail) [12]; and socioeconomic status (SES) (determined using the Index of Relative Socioeconomic Disadvantage (IRSD), with higher values indicating lower socioeconomic disadvantage) [14]. The Charlson Comorbidity Index (CCI) was also used to account for comorbidity burden. We also captured data on patients discharged over the weekend and after hours (after 1700 h).

Other factors assessed were the number of emergency department (ED) visits in the previous six months and hospital readmissions within the past year [15]. Laboratory variables obtained on admission from the EMRs included haemoglobin levels; white blood cell (WBC) counts; neutrophil and lymphocyte counts; and creatinine, C-reactive protein (C-RP), albumin, and sodium levels. The neutrophil-to-lymphocyte ratio (NLR) [16], which has recently been found to be related to clinical outcomes in hospitalised patients, was determined by dividing the absolute neutrophil and lymphocyte counts on admission. Polypharmacy was defined as the use of ≥5 medications on admission [17].

In this study, two traditional models were used as baseline comparators: (1) a stepwise backward logistic regression model, which represents an optimised version of traditional logistic regression by retaining only statistically significant predictors based on backward elimination, and (2) the LACE index, a widely used rule-based scoring tool for predicting hospital readmissions. These were compared against four machine learning models (Least Absolute Shrinkage and Selection Operator (LASSO), random forest (RF), Extreme Gradient Boosting (XGBoost), and artificial neural networks (ANNs)) to assess relative predictive performance.

#### LACE Index

The LACE index is a continuous score ranging from 0 to 19 used to quantify the risk of death or unplanned readmission within 30 days of discharge [18]. The LACE index is calculated based on the following variables: the length of stay (LOS), acute or urgent admission, the CCI, and the number of ED visits in the previous six months [18]. A score ≥ 10 is associated with a high risk of readmission, with an expected probability ranging from 12.2% to 47% according to previous studies [19].

### 2.2. Statistical Analysis

Data normality was assessed through a visual inspection of histograms. The LOS, HFRS, urea, and C-RP were log- transformed because of marked skewness. The proportion of missing data for each variable is presented in Appendix A. Missing data were handled using simple imputation, with the median imputed for continuous variables and the mode for categorical variables. The demographic characteristics of patients with and without 30-day readmissions were compared. Categorical variables were summarised as proportions, while continuous variables were summarised as means with standard deviations (SDs) or medians with interquartile ranges (IQRs), as appropriate. Comparisons of continuous variables between groups were performed using *t*-tests for normally distributed data or Wilcoxon rank-sum tests for non-normally distributed data. Categorical variables were compared using chi-square tests.

The study sample was randomly divided into training and testing datasets in a ratio of 80:20. To assess the predictive performance of six different models (including four ML models) for predicting 30-day readmissions, we performed 10-fold cross-validation on each model. The models compared were as follows: stepwise logistic regression, LACE index, LASSO regression, RF, XGBoost, and ANN. The primary evaluation metric used was the area under the receiver operating characteristic curve (AUC), with 95% confidence intervals (CIs) calculated for each model’s AUC.

The stepwise logistic regression model used the backwards selection procedure with all available predictors (see Appendix A initially considered and then non-significant variables (*p* > 0.05) being sequentially removed. Odds ratios (ORs) with 95% confidence intervals (CIs) were reported for included variables. An additional logistic regression model that included only the LACE index as a predictor was also assessed. Model performance was evaluated using the AUC and the Brier score, while model calibration was assessed using the Hosmer–Lemeshow goodness-of-fit test.

### 2.3. Machine Learning Prediction Models

We compared the following ML algorithms against the stepwise logistic regression model and the LACE index in predicting 30-day readmissions:

LASSO: This regression-based approach, akin to logistic regression, uses maximum likelihood estimation to identify solutions. However, LASSO incorporates a penalised maximum likelihood estimate, encouraging some parameters to be set to zero, effectively performing variable selection by retaining only non-zero coefficients [20]. For this model, continuous variables were standardised using z-score normalisation. To identify the optimal regularisation strength (λ), 10-fold stratified cross-validation was performed using grid search across a logarithmically spaced range of inverse regularisation values (C = 10^−4^ to 10^2^). The logistic regression model was fitted using the saga solver, which is appropriate for L1-penalised models. The primary optimisation criterion was the AUC, with the mean AUC across folds used to select the optimal hyperparameter.

A fixed random seed (random_state = 42) ensured reproducibility across all model training and validation procedures. Model performance was subsequently evaluated on a held-out test set (20% of the data) using the AUC and the Brier score, and model calibration was assessed using the Hosmer–Lemeshow goodness-of-fit test.

RF: This ensemble method constructs multiple decision trees based on bootstrapped samples of the training data. At each split, a random subset of features is selected for prediction. The model aggregates predictions from multiple trees to improve accuracy and prevent overfitting [21]. Hyperparameter tuning was performed using 10-fold stratified cross-validation and a grid search approach. The tuning grid included the following: n_estimators [100, 200, 300], max_depth [5, 10, 15], min_samples_split [2, 5, 10], and min_samples_leaf [1, 2, 4]. The random seed was fixed at 42 to ensure reproducibility. Model performance was assessed via the mean AUC across 10 folds, with 95% confidence intervals, and model calibration was evaluated using the Hosmer–Lemeshow test.

XGBoost: This method uses gradient-boosted decision trees, where trees are iteratively grown based on the outcomes of previously constructed trees. Each new tree is aimed at correcting the misclassifications made by earlier trees, giving more emphasis to difficult cases [22]. The model uses regularisation to prevent overfitting and is capable of capturing both linear and non-linear relationships in the data, which makes it particularly effective for predictive tasks. Hyperparameter tuning was performed using RandomizedSearchCV with 10-fold cross-validation, optimising for the AUC. The key parameters tuned included n_estimators (100–1000), max_depth (3–10), learning_rate (0.01–0.3), subsample and colsample_bytree (0.5–1.0), gamma (0–5), min_child_weight (1–10), and scale_pos_weight (1–5) to address class imbalance. One hundred parameter combinations were tested using a fixed random seed (42). The best-performing model was evaluated on a held-out test set. Model performance was assessed using the AUC and Brier score, and calibration was evaluated using the Hosmer–Lemeshow goodness-of-fit test.

ANN: Inspired by biological neural networks, ANNs consist of nodes that communicate through weighted connections. A feedforward neural network processes input features and their targets to find optimal hyperplanes that separate classes in multi-dimensional space. Unlike logistic regression, ANNs output class associations rather than probabilities [23]. Hyperparameter tuning was performed using RandomizedSearchCV with 10-fold cross-validation. Parameters included the following: the number of hidden layers (1 or 2), units per layer (16, 32, 64, 128), dropout rate (0.2–0.5), learning rate (0.0001–0.01, log scale), batch size (16, 32, 64), and epochs (50 or 100). A fixed random_state = 42 ensured reproducibility. The model used the Adam optimiser with binary cross-entropy loss and was evaluated using the AUC. Calibration was assessed using the Hosmer–Lemeshow goodness-of-fit test.

### 2.4. Sensitivity Analysis

We performed a sensitivity analysis by imputing missing data using the Multivariate Imputation by Chained Equations (MICE) approach via IterativeImputer from scikit-learn. A single imputed dataset was generated and used to refit all predictive models, including logistic regression, random forest, LASSO, XGBoost, and neural networks. Model performance, including the AUC, was evaluated on the imputed data to assess the robustness of the findings to missing data.

### 2.5. Comparison of Models

The performance of the regression and ML models was evaluated using multiple metrics, reflecting their ability to distinguish between cases and non-cases. These metrics included the concordance statistic (C-statistic) and the Brier score. The Brier score quantifies overall model performance by measuring the disagreement between observed outcomes and predicted probabilities, with lower values indicating better predictive accuracy. A Brier score of <0.25 is considered indicative of good performance.

To assess whether differences in AUCs between the models were statistically significant, pairwise comparisons were performed using bootstrapped confidence intervals (CIs) for the mean differences in AUCs. Bootstrapping is a resampling technique that creates multiple new datasets by sampling with replacement from the original data. In this study, the bootstrapping procedure involved drawing 1000 resampled datasets for each model comparison. This approach provides stable estimates of AUC variability and ensures the calculation of reliable confidence intervals, which are narrower due to the large number of resamples.

The DeLong test for AUC comparison was applied to determine statistical significance between pairs of models [24]. All statistical tests were two-sided, and a *p* value of <0.05 was considered statistically significant.

Statistical analyses were performed using STATA software version 18.0 and Python version 3.12.4 with the following libraries: Numpy, pandas, scikit-learn, statsmodels, XGboost, and Tensorflow.

## 3. Results

During the study period, 5688 patients were admitted under general medicine, with 1341 (23.8%) experiencing a readmission within 30 days of discharge. We excluded 317 patients who had multiple readmissions (>1) to ensure only a single outcome per patient, facilitating model consistency and avoiding outcome dependence. The final cohort included 5371 patients, including 1024 (19.1%) readmissions (Figure 1). The mean (SD) age of the cohort was 68.9 (19.1) years (range 18–102 years), and 46.6% were male. The mean (SD) Charlson index was 1.5 (2.1), and 2134 patients (39.7%) were classified as frail according to the HFRS (HFRS ≥ 5).

The characteristics of patients, stratified by 30-day readmission status, are presented in Table 1. Compared to patients who did not experience 30-day readmission, readmitted patients were significantly older and frail, were less likely to be living at home, had a lower SES, a greater comorbidity burden, and higher number of prior hospital admissions (*p* < 0.05). Additionally, readmitted patients had higher rates of hypertension, diabetes, congestive heart failure (CHF), and chronic lung disease compared to non-readmitted patients. Inflammation was also greater among these patients, evidenced by an elevated NLR alongside lower albumin levels. The median length of stay (LOS) was significantly longer for readmitted patients, who were also more likely to be on polypharmacy at admission. Readmitted patients also had a higher mean LACE score than non-readmitted patients (*p* < 0.001).

### 3.1. Predictive Ability of LACE Index

The mean (SD) LACE index was 9.1 (3.0), with 2206 (41.1%) patients classified as high-risk (LACE score ≥ 10). A higher LACE index score was significantly associated with an increased risk of readmissions (OR 1.13, 95% 1.10–1.15, *p* < 0.001). Patients in the high-risk LACE category had an 87% higher risk of 30-day readmission compared to those in the low-risk LACE category (OR 1.87 95% CI 1.63–2.15, *p* < 0.001). The AUCROC for the LACE index was 0.61 (95% CI 0.58–0.63) (Figure 2). The Brier score was 0.15 (95% CI 0.14–0.16), indicating reasonable predictive accuracy, and model calibration was acceptable, as shown with the Hosmer–Lemeshow goodness-of-fit test, with χ^2^ = 13.76, and *p* = 0.0883.

### 3.2. Stepwise Backwards Logistic Regression Model

The stepwise logistic regression model identified ten variables significantly associated with 30-day readmissions. A history of CHF, alcohol abuse, nursing home residency, and prior hospital admissions were associated with a significantly higher odds of readmission within 30 days of discharge (Table 2). The mean AUCROC from the 10-fold cross-validation was 0.62 (95% CI 0.60–0.65) (Table 2 and Figure 2). The Brier score was 0.15 (95% CI 0.14–0.16), indicating a reasonable level of predictive accuracy. Model calibration was acceptable, with a Hosmer–Lemeshow goodness-of-fit test yielding χ^2^ = 12.59 and *p* = 0.1266.

### 3.3. Machine Learning Models

Among the ML models, LASSO regression selected 21 variables (Appendix A) for model inclusion and demonstrated the best predictive ability with an AUCROC of 0.63 (95% CI 0.58–0.68), with the Hosmer–Lemeshow goodness-of-fit test yielding χ^2^ = 13.74 and *p* = 0.185.

The random forest model yielded a 10-fold cross-validated mean AUCROC of 0.60 (95% CI: 0.58–0.68), indicating moderate discriminative performance. However, model calibration was suboptimal, with a statistically significant Hosmer–Lemeshow test result (χ^2^ = 17.32; *p* = 0.0269), suggesting discrepancies between the predicted and observed event probabilities. Calibration improved modestly with Isotonic regression (χ^2^ = 16.76; *p* = 0.0327), though the result remained statistically significant, indicating residual miscalibration.

XGBoost had an AUC of 0.55 (95% CI 0.53–0.56), with the Hosmer-Lemeshow goodness-of-fit test yielding χ^2^ = 608.90 and *p* < 0.05. After recalibration using Platt scaling, the XGBoost model demonstrated good calibration, with Hosmer–Lemeshow test results of χ^2^ = 4.70 and *p* = 0.7886, indicating no significant difference between the predicted and observed probabilities across deciles.

The ANN yielded an AUC of 0.60 (95% CI 0.57–0.63) with reasonable calibration, with the Hosmer–Lemeshow goodness-of-fit test yielding χ^2^ = 14.77 and *p* = 0.0639 (Figure 2). The Brier scores for LASSO, RF, XGBoost, and the ANN were 0.15 (95% CI 0.14–0.17), 0.15 (95% CI 0.14–0.16), 0.19 (95% CI 0.18–0.20), and 0.15 (95% CI 0.15–0.18), respectively, indicating reasonably good predictive accuracy.

### 3.4. Sensitivity Analysis

Sensitivity analyses were performed using a dataset imputed via multiple imputation to assess the robustness of the model findings. Overall, the results were consistent with the primary analysis. The stepwise backward logistic regression model demonstrated an AUC of 0.63 (95% CI: 0.62–0.63), with a Hosmer–Lemeshow χ^2^ statistic of 13.56 (*p* = 0.094), indicating good calibration. The LACE index yielded an AUC of 0.61 (95% CI: 0.58–0.63) and a Hosmer–Lemeshow χ^2^ of 13.79 (*p* = 0.087). LASSO regression produced an AUC of 0.63 (95% CI: 0.59–0.68) with a Hosmer–Lemeshow χ^2^ of 15.62 (*p* = 0.048), suggesting marginal calibration. The random forest model achieved an AUC of 0.62 (95% CI: 0.58–0.66) and excellent calibration (Hosmer–Lemeshow χ^2^ = 2.05, *p* = 0.979). XGBoost resulted in a lower AUC of 0.57 (95% CI: 0.54–0.60) with a Hosmer–Lemeshow χ^2^ of 15.5 (*p* = 0.050). Lastly, the artificial neural network (ANN) model demonstrated an AUC of 0.60 (95% CI: 0.58–0.62) and acceptable calibration (Hosmer–Lemeshow χ^2^ = 9.75, *p* = 0.283).

### 3.5. Pairwise Comparison of AUCs (Figure 3)

Stepwise Logistic Regression vs. LACE Index: No significant difference (Mean AUC Difference: 0.01; 95% CI: −0.026–0.046, *p* > 0.05).Stepwise Logistic Regression vs. LASSO: No significant difference (Mean AUC Difference: −0.01; 95% CI: −0.066–0.047, *p* > 0.05).Stepwise Logistic Regression vs. Random Forest: No significant difference (Mean AUC Difference: 0.02; 95% CI: −0.010–0.049, *p* > 0.05).Stepwise Logistic Regression vs. XGBoost: Significant difference (Mean AUC Difference: 0.07; 95% CI: 0.040–0.099, *p* < 0.05).Stepwise Logistic Regression vs. Neural Network: No significant difference (Mean AUC Difference: 0.02; 95% CI: −0.020–0.060, *p* > 0.05).LACE Index vs. LASSO: No significant difference (Mean AUC Difference: 0.02; 95% CI: −0.076–0.036, *p* > 0.05).

**Figure 3 healthcare-13-01223-f003:**
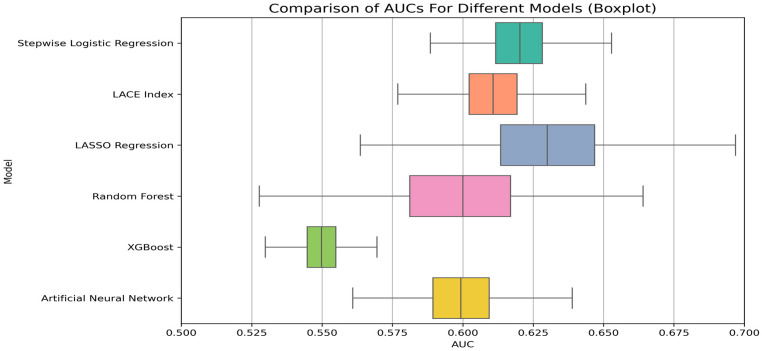
A comparison of the area under the curve (AUC) of different models.

### 3.6. Statistical Significance

The DeLong test showed significant differences between stepwise logistic regression and XGBoost (*p* < 0.05), while no significant differences were found between stepwise logistic regression and the LACE index, stepwise logistic regression and LASSO, stepwise logistic regression and the RF, stepwise logistic regression and a neural network, or between the LACE index and LASSO (*p* > 0.05) (Figure 3).

## 4. Discussion

In this study, 19.1% of general medical patients discharged alive were readmitted within 30 days. Readmitted patients were generally older and frailer and had more comorbidities, lower socioeconomic status (SES), and a higher number of prior hospitalisations in the preceding year. Despite incorporating multiple patient characteristics, predictive models demonstrated only modest accuracy in forecasting 30-day readmissions. Stepwise logistic regression performed similarly to LASSO regression and marginally outperformed the LACE index, while other machine learning (ML) models, including random forest and XGBoost, showed inferior predictive ability.

### 4.1. Comparison with Previous Studies

The 30-day readmission rate in our study was 19.1%, aligning with the findings from previous studies involving adult general medical patients [7]. In contrast, significantly lower readmission rates have been reported in younger populations. A recent study examining paediatric surgical patients found 30-day readmission rates ranging from 0.57% to 0.99% over a three-year period, highlighting substantial differences in readmission risk between adult and paediatric cohorts [25].

Our findings align with prior research [26,27] that has also reported the modest predictive performance of various models for hospital readmissions, including those restricted to specific patient populations. For example, one study focusing on patients admitted with COPD exacerbation found that the HOSPITAL score using logistic regression and an RF model had AUC values of 0.63 (95% CI: 0.59–0.67) and 0.69 (95% CI: 0.66–0.73), respectively, for predicting 90-day readmissions. Similarly, a study of 272,778 heart failure patients found that logistic regression with LASSO and neural networks had comparable, albeit modest, predictive performance for 30-day readmissions (AUC = 0.64 for both models). These findings underscore the challenge of accurately predicting 30-day readmissions with the currently available models. A variety of factors are involved in hospital readmissions, and many of them may be unpredictable, for example, social factors which are difficult to quantify and may not be readily available in a database [28]. Similarly, the lack of information on factors such as functional status [29] and access to post-discharge follow-up care [30], which have been linked to unplanned readmissions, may be responsible for the poor performance of current models.

### 4.2. Performance of Machine Learning vs. Logistic Regression

In our study, traditional logistic regression demonstrated fair predictive power (AUC = 0.62) which was comparable to LASSO regression and superior to other ML models. These findings contradict a previous study [5] that reported the superior performance of ML models (XGBoost and random forest) over logistic regression in predicting readmissions among older heart failure patients. In that study, the AUCs for XGBoost and RF were 0.80 (95% CI: 0.73–0.87) and 0.77 (95% CI: 0.70–0.84), respectively, which were notably higher than the values observed in our study (0.55 (95% CI: 0.53–0.56) for XGBoost and 0.60 (95% CI: 0.58–0.68) for random forest). A key factor contributing to this discrepancy may be differences in study populations. While the previous study focused on a homogeneous cohort of older heart failure patients, our study included patients with a broader range of diagnoses and age groups, potentially leading to lower model performance. Additionally, the previous study incorporated a more extensive set of predictor variables, including detailed laboratory parameters, medication data, and measures of physical activity, which may have enhanced model accuracy and resulted in higher AUC values. Beyond differences in study populations and predictor availability, the previous study employed a structured approach to feature engineering, data preprocessing, and feature selection, which may have further optimised model performance. Their methodology included handling missing data through MICE imputation, balancing the dataset via under-sampling, and applying LASSO for feature selection. These steps likely enhanced the predictive power of their models by refining the feature set and improving data quality, whereas our study utilised a predefined set of clinically relevant variables without extensive feature selection techniques.

Our results, however, align with the findings from a systematic review of 71 studies comparing ML models (classification trees, random forests, artificial neural networks, support vector machines) with logistic regression for various clinical outcomes [11]. That study found that the difference in logit(AUC) between logistic regression and ML models was 0.00 (95% CI: −0.18 to 0.18), indicating no clear advantage of ML over traditional logistic regression.

One possible explanation for the lack of ML superiority in predicting readmissions is the limited number of predictor variables. ML models perform best when handling large datasets with a high number of predictors, allowing them to capture complex, non-linear relationships [31,32]. When the number of predictors is small, logistic regression often performs equally well or better.

### 4.3. Performance of LACE Index

Our study also found that the LACE index had only modest predictive ability in general medical patients and was inferior to both LASSO regression and traditional logistic regression. This is consistent with previous studies [5,33] that reported the limited prediction accuracy of the LACE index across different patient populations. For instance, a Canadian study of 26,000 medical patients from six hospitals found that 12.6% of patients were readmitted within 30 days. In that study, while 34% of patients had a high-risk LACE index (>10), it correctly identified only 51.7% of readmitted patients [34].

### 4.4. Key Predictors of Readmissions

Consistent with previous studies [35,36], our study found a relatively high rate of 30-day readmissions among general medical patients. Furthermore, we identified the key predictors of readmission, including a history of heart failure, alcohol abuse, discharge after hours, and a higher number of prior hospitalisations—findings that align with prior research [37,38,39].

### 4.5. Limitations

This study has several limitations. This study was conducted in a single tertiary care hospital, limiting generalisability to other healthcare settings, with different patient populations, care models, and resource availability. We excluded patients with multiple 30-day readmissions to maintain a single outcome per patient; however, this may introduce bias, as these patients may have distinct clinical or social characteristics influencing readmission risk. We were unable to collect data on patients’ functional status, which is a significant factor influencing hospital readmissions [29]. Our models were developed and internally validated using 10-fold cross-validation on the same dataset, but no external validation was performed. Without testing on an independent cohort, the generalisability and transportability of our models to other settings remain uncertain.

Additionally, we employed stepwise logistic regression for variable selection in one of our models. While commonly used, stepwise methods have well-documented limitations, including reliance on arbitrary significance thresholds, an increased risk of overfitting, and model instability—particularly in the presence of multicollinearity or modest sample sizes. These approaches may also lead to biassed estimates and understate the true variability in predictor selection. To address this, we supplemented stepwise regression with LASSO regularisation, which offers a more robust and data-driven method of variable selection by penalising model complexity and reducing overfitting. The consistency of key predictors across both approaches strengthens the reliability of our findings.

ML models typically require large datasets with numerous predictors to achieve stable and reliable performance. The relatively small number of predictors in our study may have hindered the full potential of ML techniques. Further validation using both retrospective and prospective cohorts with additional clinical and social determinants of health is required to enhance the predictive performance of existing models.

## 5. Conclusions

This study found that a substantial proportion of general medical patients experience 30-day readmissions. The predictive performance of traditional logistic regression models was only modest but was comparable or superior to that of ML methods. Future research should explore incorporating additional variables, including functional status and social factors, to improve model performance.

## Figures and Tables

**Figure 1 healthcare-13-01223-f001:**
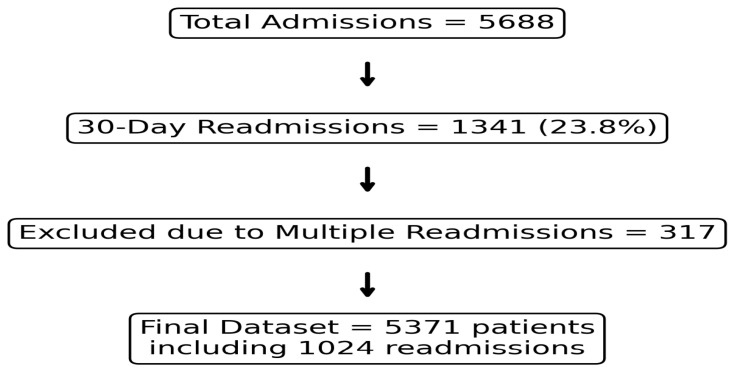
Study flow diagram.

**Figure 2 healthcare-13-01223-f002:**
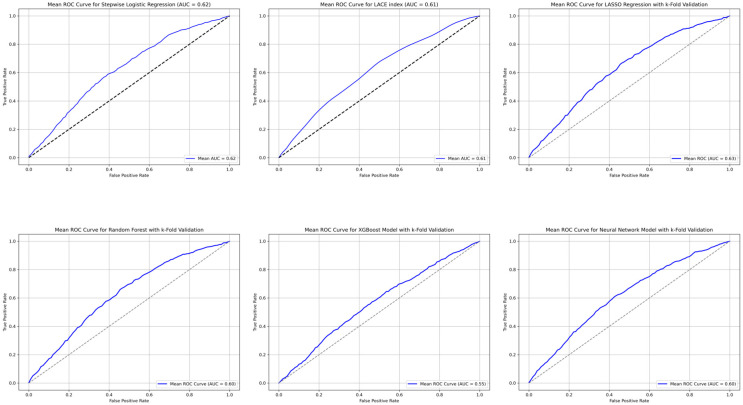
Receiver operating characteristic (ROC) curves for different models.

**Table 1 healthcare-13-01223-t001:** Characteristics of readmitted versus non-readmitted patients.

Variable	Not Readmitted	Readmitted Within 30 Days	*p* Value
N (%)	4347 (80.9)	1024 (19.1)	
Mean age (SD)	68.5 (19.4)	70.8 (17.4)	<0.001
Age group n (%)			
<40	489 (11.2)	89 (8.6)	0.009
40–59	659 (15.1)	129 (12.6)	
60–79	1725 (39.6)	432 (42.1)	
>80	1474 (33.9)	374 (36.5)	
Male sex n (%)	2016 (47.4)	484 (47.3)	0.608
Living at home n (%)	3983 (91.6)	869 (84.8)	<0.001
Indigenous n (%)	92 (2.1)	25 (2.4)	0.522
Frail n (%)	1667 (38.3)	467 (45.6)	0.001
Mean HFRS (SD)	5.5 (4.4)	6.3 (5.2)	<0.001
Mean ED visits last 6 months (SD)	1.2 (2.9))	1.9 (2.8)	<0.001
Mean number of admissions in last year (SD)	1.0 (2.0)	1.5 (2.4)	<0.001
Mean IRSD (SD)	998.2 (58.6)	993.2 (62.3)	0.015
Mean Charlson index (SD)	1.4 (2.0)	1.9 (2.3)	<0.001
Hypertension n (%)	354 (8.1)	113 (11.0)	0.003
Diabetes n (%)	1121 (25.7)	307 (29.9)	0.006
CHF n (%)	811 (18.6)	271 (26.4)	<0.001
CAD n (%)	302 (6.9)	85 (8.3)	0.132
Chronic lung disease n (%)	907 (20.8)	250 (24.4)	0.013
Stroke n (%)	78 (1.8)	17 (1.6)	0.769
Mean haemoglobin level (SD)	127.5 (20.9)	123.2 (22.3)	<0.001
Mean WBC count (SD)	10.3 (10.4)	9.9 (5.5)	0.311
Mean platelet count (SD)	243.5 (94.6)	246.3 (109.7)	0.420
Mean NLR (SD)	8.3 (9.9)	9.2 (12.1)	0.012
Mean C-RP level (SD)	50.2 (76.7)	49.9 (73.6)	0.900
Mean urea level (SD)	7.8 (5.6)	8.6 (6.7)	<0.001
Mean creatinine level (SD)	97.2 (68.8)	100.9 (67.9)	0.117
Mean sodium level (SD)	137.8 (4.9)	137.4 (5.3)	0.023
Mean albumin level (SD)	32.9 (5.6)	32.2 (5.7)	<0.001
Median LOS (IQR)	3.2 (1.8, 6.1)	4.1 (2.1, 8.3)	<0.001
Discharged over the weekend n (%)	760 (17.4)	161 (15.7)	0.179
Discharged after hours n (%)	812 (18.6)	223 (21.7)	0.024
Polypharmacy n (%)	2599 (59.7)	655 (63.9)	0.014
Mean LACE score (SD)	8.8 (2.9)	10.0 (3.0)	<0.001
High-risk LACE n (%)	1657 (38.1)	549 (53.6)	<0.001

SD, standard deviation; HFRS, Hospital Frailty Risk Score; ED, emergency department; IRSD, Index of Relative Socioeconomic Disadvantage; CHF, congestive heart failure; CAD, coronary artery disease; WBC, white blood cell; NLR, neutrophil–lymphocyte ratio; C-RP, C-reactive protein; LOS, length of hospital stay; LACE, index calculated from Length of hospital stay, Admission type, Charlson index and Emergency department visits in the previous 6 months.

**Table 2 healthcare-13-01223-t002:** Variables associated with 30-day readmission after using stepwise backward logistic regression model.

Variable	OR	95% CI	*p* Value
Charlson index	1.07	1.03–1.09	<0.001
HFRS	1.02	1.01–1.04	0.008
Hospital admissions in previous year	1.08	1.04–1.11	<0.001
Congestive heart failure	1.25	1.04–1.40	<0.001
Alcohol abuse	1.53	1.20–1.65	<0.001
Residence nursing home	1.44	1.22–1.51	0.041
Urea levels	1.03	1.01–1.05	0.008
Creatinine levels	0.99	0.98–0.99	0.008
Haemoglobin levels	0.99	0.98–0.99	0.001
Sodium levels	0.98	0.97–0.99	<0.001

OR, odds ratio; CI, confidence interval; HFRS, Hospital Frailty Risk Score.

## Data Availability

The data presented in this study are available on request from the corresponding author if permission is granted by the ethics committee.

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
