# Peer review of "Performance of Machine Learning Models in Predicting 30-Day General Medicine Readmissions Compared to Traditional Approaches in Australian Hospital Setting"

_healthcare, 2025, doi:10.3390/healthcare13111223_

Round 1

Reviewer 1 Report

Comments and Suggestions for Authors

The authors compared the predictive performance of nonlinear machine learning (ML) models with stepwise logistic regression (LR) and the LACE index for predicting 30-day general medicine readmissions in a tertiary hospital in Australia over a one-year period. They found that about one in five general medicine patients were readmitted within 30 days. Traditional LR models performed as well or better than ML models in predicting readmission risk.

Given the global burden of hospital readmissions and the increasing use of ML in healthcare, the topic is timely and highly relevant. The study is well designed and written. The methodology is described with reasonable clarity, including data sources, variable selection, and validation approach.

I have no major comments. My comments and suggestions for improvement are as follows:

  1. Abstract – Any abbreviation that appears in the abstract should be expanded the first time it is mentioned. Please expand the abbreviations LACE and LASSO.
  2. The authors state that the Southern Adelaide Clinical Human Research Ethics Committee (SAC HREC) has classified this study as not requiring ethical approval. In this case, you should provide the reference number and date of the document indicating that ethical approval is not required.
  3. The inclusion and exclusion criteria are not clearly defined. E.g. whether the pediatric patients were excluded or the patients with missing data. Please revise.
  4. Study outcomes – the authors should clearly define the primary and secondary outcomes of the study and present them in a separate paragraph in the methodology.
  5. The authors only briefly mentioned how they dealt with missing data. In my opinion, this is not described in sufficient detail. The imputation methods and the exclusion criteria need to be explained in more detail.
  6. The authors have not specified the baseline model (traditional methods) sufficiently. It is not clear from the methodology whether the logistic regression is a base comparison model or whether it is an optimized traditional model. Please clarify this!
  7. The model performance metrics lack confidence intervals that could help to understand the variability.
  8. In the discussion (paragraph - comparison with previous studies), the authors should add a reference to the difference with the pediatric population. The most recent study clearly showed a significantly lower incidence of pediatric readmissions within 30 days. The overall readmission rate is between 0.57% and 0.99%. Please add a comment and reference (doi: 10.1080/00015458.2021.1927657).
  9. The models used in this study have not been externally validated. This may limit the generalizability of the data obtained beyond the hospital studied. Future work should include multi-site validation or external validation, which should be addressed under the limitations of the study.

Author Response

The authors compared the predictive performance of nonlinear machine learning (ML) models with stepwise logistic regression (LR) and the LACE index for predicting 30-day general medicine readmissions in a tertiary hospital in Australia over a one-year period. They found that about one in five general medicine patients were readmitted within 30 days. Traditional LR models performed as well or better than ML models in predicting readmission risk.

Given the global burden of hospital readmissions and the increasing use of ML in healthcare, the topic is timely and highly relevant. The study is well designed and written. The methodology is described with reasonable clarity, including data sources, variable selection, and validation approach.

I have no major comments. My comments and suggestions for improvement are as follows:

  1. Abstract – Any abbreviation that appears in the abstract should be expanded the first time it is mentioned. Please expand the abbreviations LACE and LASSO. Please refer to page 1 under Abstract section.

Response: We thank reviewer for the comments. We have expanded the abbreviations as per reviewer’s suggestion.

“..the LACE index (Length of stay, Admission type, Comorbidity index, and recent Emergency department visits)..”

“Extreme Gradient Boosting (XGBoost)”

“Least Absolute Shrinkage and Selection Operator (LASSO)”

“Artificial Neural Networks (ANN)”

2. The authors state that the Southern Adelaide Clinical Human Research Ethics Committee (SAC HREC) has classified this study as not requiring ethical approval. In this case, you should provide the reference number and date of the document indicating that ethical approval is not required.

Response: This has now been included in the manuscript. Please refer to page 3.

“The Southern Adelaide Clinical Human Research Ethics Committee (SAC HREC) deemed this study exempt from requiring ethical approval (reference number 4774 dated 14 November 2023).”

3. The inclusion and exclusion criteria are not clearly defined. E.g. whether the pediatric patients were excluded or the patients with missing data. Please revise.

Response: We have now clarified the inclusion and exclusion criteria. Please refer to page 3.

“This retrospective study included all adult patients (aged ≥18 years) admitted under the General Medicine Department at Flinders Medical Centre (FMC) between 1 July 2022 and 30 June 2023. Patients were eligible if they were discharged alive and data were obtained from the electronic medical records (EMR) database. Readmission status was determined based on any unplanned hospital readmission within 30-days of discharge.”

4. Study outcomes – the authors should clearly define the primary and secondary outcomes of the study and present them in a separate paragraph in the methodology.

Response: We have now clearly defined primary and secondary outcomes in the methodology section. Please refer to page no 3.

“The primary outcome was all-cause unplanned hospital readmission within 30 days of discharge from a general medicine admission. Secondary outcomes included identification of significant predictors of readmission and comparison of predictive performance between stepwise logistic regression, the LACE index, and machine learning models using the area under the receiver operating characteristic curve (AUC).”

5. The authors only briefly mentioned how they dealt with missing data. In my opinion, this is not described in sufficient detail. The imputation methods and the exclusion criteria need to be explained in more detail.

Response: The missing data has now been described. A new table has been added which describes the proportion of missing data per variable. In addition, as per other reviewer’s suggestion a sensitivity analysis has been performed using multiple imputation. Please refer to pages 4 and 6.

“The proportion of missing data for each variable is presented in Supplementary Table 1. Missing data were handled using simple imputation, with the median imputed for continuous variables and the mode for categorical variables.”

“Sensitivity analysis

We performed a sensitivity analysis by imputing missing data using the Multivariate Imputation by Chained Equations (MICE) approach via IterativeImputer from scikit-learn. A single imputed dataset was generated and used to refit all predictive models, including logistic regression, random forest, LASSO, XGBoost, and neural networks. Model performance, including AUC, was evaluated on the imputed data to assess the robustness of findings to missing data.”

6. The authors have not specified the baseline model (traditional methods) sufficiently. It is not clear from the methodology whether the logistic regression is a base comparison model or whether it is an optimized traditional model. Please clarify this!

Response: We have clarified the description of our baseline models in the Methods section. Specifically, we used two traditional models for comparison: (1) a stepwise backward logistic regression model, which represents an optimized traditional approach by iteratively removing non-significant predictors to improve model performance and reduce overfitting, and (2) the LACE index, a validated rule-based scoring system. These models served as baseline comparators against which the performance of the machine learning models (LASSO, Random Forest, XGBoost, and ANN) was evaluated. Please refer to page no 3-4.

In this study, two traditional models were used as baseline comparators: (1) a stepwise backward logistic regression model, which represents an optimised version of traditional logistic regression by retaining only statistically significant predictors based on backward elimination, and (2) the LACE index, a widely used rule-based scoring tool for predicting hospital readmissions. These were compared against four machine learning models Least Absolute Shrinkage and Selection Operator (LASSO, Random Forest, Extreme Gradient Boosting (XGBoost), and Artificial Neural Networks (ANN)) to assess relative predictive performance.”

7. The model performance metrics lack confidence intervals that could help to understand the variability.

Response: We have included 95% confidence intervals (CIs) for all model performance metrics in the Results section (pages 10–12) to better represent the variability and precision of our estimates.

“The area under the ROC curve (AUCROC) for the LACE index was 0.61 (95% CI 0.58-0.63)”

“The mean AUCROC from the 10-fold cross-validation was 0.62 (95% CI 0.60-0.65)”

“Among the ML models, LASSO regression selected 21 variables (Figure S1) for model inclusion and demonstrated the best predictive ability with an AUCROC of 0.63 (95% CI 0.58-0.68)”

“The Random Forest model yielded a 10-fold cross-validated mean AUCROC of 0.60 (95% CI: 0.58–0.68),”

“XGBoost had an AUC of 0.55 (95% CI 0.53-0.56)”

“and the ANN yielded an AUC of 0.60 (95% CI 0.57-0.63)”

8.  In the discussion (paragraph - comparison with previous studies), the authors should add a reference to the difference with the pediatric population. The most recent study clearly showed a significantly lower incidence of pediatric readmissions within 30 days. The overall readmission rate is between 0.57% and 0.99%. Please add a comment and reference (doi: 10.1080/00015458.2021.1927657).

Response: We have updated the discussion section to include a comparison with paediatric populations and referenced the recent study (doi: 10.1080/00015458.2021.1927657) demonstrating significantly lower 30-day readmission rates in paediatric surgical patients. This addition helps contextualise our findings within broader age-related differences in readmission risk. Please refer to page no 13.

“The 30-day readmission rate in our study was 19.1%, aligning with findings from previous studies involving adult general medical patients [24]. In contrast, significantly lower readmission rates have been reported in younger populations. A recent study examining paediatric surgical patients found 30-day readmission rates ranging from 0.57% to 0.99% over a three-year period, highlighting substantial differences in readmission risk between adult and paediatric cohorts [25].”

9. The models used in this study have not been externally validated. This may limit the generalizability of the data obtained beyond the hospital studied. Future work should include multi-site validation or external validation, which should be addressed under the limitations of the study.

Response: Thank you for highlighting this important limitation. We agree that our models, while internally validated using 10-fold cross-validation, have not undergone external validation. As such, their generalisability and transportability to other hospitals or healthcare systems remain uncertain. We have revised the Limitations section to explicitly acknowledge this and added a statement emphasising the need for future multi-site or external validation to confirm model robustness across diverse clinical settings. Please refer to page no 15.

“Our models were developed and internally validated using 10-fold cross-validation on the same dataset, but no external validation was performed. Without testing on an independent cohort, the generalisability and transportability of our models to other settings remain uncertain.”

References:

  1. Jukić, M.; Antišić, J.; Pogorelić, Z. Incidence and causes of 30-day readmission rate from discharge as an indicator of quality care in pediatric surgery. Acta Chir. Belg. 2023, 123, 26-30, doi:10.1080/00015458.2021.1927657.

Reviewer 2 Report

Comments and Suggestions for Authors

Dear authors, 

Thank you so much for the opportunity to review your work, below few suggestions you might consider or clarify:

Erroneous AUC differences and CIs

In the pairwise comparisons, the reported “Mean AUC Difference” values of –0.3499 for LR vs. Random Forest and XGBoost are internally inconsistent with the AUCs themselves (0.62 vs. 0.60 and 0.55, respectively) . A true difference of 0.62–0.60=0.02, not 0.35—and confidence intervals should span around that. This error critically undermines all statistical significance claims for model comparisons.

Inconsistent reporting of model performance in the Discussion

The Discussion states Random Forest AUC=0.50 (95% CI 0.59–0.62), which conflicts both with the Results (AUC=0.60, CI 0.59–0.62) and with logical CI placement (lower bound exceeds point estimate) 
. Such misreporting suggests the authors did not carefully cross-check their numbers.

Table 2 duplication and mismatch with text

Table 2 lists “Sodium levels” twice, with slightly different CIs: 0.98 (0.98–0.99) and 0.98 (0.97–0.99) 

.

The text indicates 11 predictors in the LR model, but only 9–10 appear in the table. This inconsistency obscures which covariates truly entered the final model.

Misaligned p-value thresholds

In describing Table 1 differences, the text reports p < 0.05 for the LACE score comparison, whereas Table 1 shows p < 0.001 

. This suggests careless transcription of significance levels.

Inadequate imputation strategy

Simple median/mode imputation for missing values can bias estimates, particularly when missingness is not completely at random. A multiple imputation approach (e.g., MICE) would better preserve variance and avoid underestimating uncertainty.

Lack of calibration assessment

The manuscript reports discrimination metrics (AUC, Brier) only. Without calibration plots or Hosmer–Lemeshow testing, it is impossible to judge whether predicted probabilities reflect observed risk—critical for clinical prediction models.

Overreliance on stepwise selection

Backwards stepwise selection inflates type I error and can yield overfit models. Modern penalisation (e.g., LASSO) or prespecified clinical predictors are preferred.

3. Detailed Recommendations for Revision

Recompute all pairwise AUC differences using the correct formula (ΔAUC = AUC₁–AUC₂), with bootstrapped CIs and DeLong test p-values. Verify consistency between Results text, Figures 2/3, and the Discussion.

Harmonise model performance reporting:

Ensure that the AUC and CI for each model in the Abstract, Results, and Discussion sections are identical.

Correct the typographical error in the Discussion (RF ≠ 0.50, should be 0.60).

Revise Table 2:

Remove duplicate sodium entries, confirm exact CIs, and include all predictors retained by the final LR model.

Clearly state the total number of variables and their selection criteria in the table legend.

Align p-values throughout. Wherever Table 1 shows p < 0.001, mirror that in the narrative.

Upgrade missing-data handling: implement multiple imputation (e.g., MICE) and report the proportion of missingness per variable. Perform sensitivity analyses to assess the impact on model performance.

Add calibration assessment:

Include calibration plots (e.g., calibration belt or LOESS) and/or Hosmer–Lemeshow statistics for each model.

Discuss any miscalibration and potential need for recalibration before clinical use.

Replace or supplement stepwise regression with a more robust variable-selection strategy. At minimum, discuss the limitations of stepwise methods and the potential for overfitting.

Detail hyperparameter tuning for ML models: specify search ranges, optimization criteria, and random seeds to ensure reproducibility.

Clarify cohort derivation:

The text states an initial cohort of 5,688 with 23.8% readmissions, then excludes 317 for multiple readmissions. Spell out reasons for exclusion and assess whether excluding multiple‐readmitted patients biases the sample.

Expand the Limitations section to discuss single-centre design, potential selection bias, and the absence of external validation.

Best wishes

Author Response

Reviewer 2

Dear authors, 

Thank you so much for the opportunity to review your work, below few suggestions you might consider or clarify:

Erroneous AUC differences and CIs

In the pairwise comparisons, the reported “Mean AUC Difference” values of –0.3499 for LR vs. Random Forest and XGBoost are internally inconsistent with the AUCs themselves (0.62 vs. 0.60 and 0.55, respectively) . A true difference of 0.62–0.60=0.02, not 0.35—and confidence intervals should span around that. This error critically undermines all statistical significance claims for model comparisons.

Response: We thank the reviewer for their comments. We have now made necessary corrections as advised by the reviewer. Please refer to page 12.

“1. Stepwise Logistic Regression vs LACE Index: No significant difference (Mean AUC Difference: 0.01, 95% CI -0.026-0.046, p>0.05).

  1. Stepwise Logistic Regression vs LASSO: No significant difference (Mean AUC Difference: -0.01, 95% CI: -0.066 – 0.047, p>0.05).
  2. Stepwise Logistic Regression vs Random Forest: No significant difference (Mean AUC Difference: 0.02, 95% CI: -0.010 - 0.049, p>0.05).
  3. Stepwise Logistic Regression vs XGBoost: Significant difference (Mean AUC Difference: 0.07, 95% CI 0.040 - 0.099, p<0.05).
  4. Stepwise Logistic Regression vs Neural Network: No significant difference (Mean AUC Difference: 0.02 , 95% CI -0.020 – 0.060, p>0.05).
  5. LACE Index vs LASSO: No significant difference (Mean AUC Difference: 0.02, 95% CI: -0.076 – 0.036, p>0.05).”

Inconsistent reporting of model performance in the Discussion

The Discussion states Random Forest AUC=0.50 (95% CI 0.59–0.62), which conflicts both with the Results (AUC=0.60, CI 0.59–0.62) and with logical CI placement (lower bound exceeds point estimate) 
. Such misreporting suggests the authors did not carefully cross-check their numbers.

Response: These errors have now been rectified. Please refer to page 13 in discussion section.

“In that study, the AUCs for XGBoost and RF were 0.80 (95% CI: 0.73–0.87) and 0.77 (95% CI: 0.70–0.84), respectively, which were notably higher than the values observed in our study (0.55 (95% CI: 0.53–0.56) for XGBoost and 0.60 (95% CI: 0.58–0.68) for Random Forest).”

Table 2 duplication and mismatch with text

Table 2 lists “Sodium levels” twice, with slightly different CIs: 0.98 (0.98–0.99) and 0.98 (0.97–0.99) 

.

Response: Errors in Table 2 have now been rectified.

The text indicates 11 predictors in the LR model, but only 9–10 appear in the table. This inconsistency obscures which covariates truly entered the final model.

Response: This error has been now corrected in results section on page 10 and Table 2.

“The stepwise logistic regression model identified ten variables significantly associated with 30-day readmissions.”

Misaligned p-value thresholds

In describing Table 1 differences, the text reports p < 0.05 for the LACE score comparison, whereas Table 1 shows p < 0.001 

. This suggests careless transcription of significance levels.

Response: The error has now been rectified. Please refer to page 8.

“Readmitted patients also had a higher mean LACE score than non-readmitted patients (p < 0.001).”

Inadequate imputation strategy

Simple median/mode imputation for missing values can bias estimates, particularly when missingness is not completely at random. A multiple imputation approach (e.g., MICE) would better preserve variance and avoid underestimating uncertainty.

Response: We have now performed multiple imputation using MICE as a sensitivity analysis as suggested by the reviewer. Please refer below for details.

Lack of calibration assessment

The manuscript reports discrimination metrics (AUC, Brier) only. Without calibration plots or Hosmer–Lemeshow testing, it is impossible to judge whether predicted probabilities reflect observed risk—critical for clinical prediction models.

Response: We have now performed model calibration with Hosmer-Lemeshow testing as per suggestions by the reviewer. Please refer below for further details.

Overreliance on stepwise selection

Backwards stepwise selection inflates type I error and can yield overfit models. Modern penalisation (e.g., LASSO) or prespecified clinical predictors are preferred.

Response: We have now added this limitation of Stepwise selection model in the limitations section.

  1. Detailed Recommendations for Revision

Recompute all pairwise AUC differences using the correct formula (ΔAUC = AUC₁–AUC₂), with bootstrapped CIs and DeLong test p-values. Verify consistency between Results text, Figures 2/3, and the Discussion.

Response: We have recomputed all pairwise AUC differences using correct formula as per reviewer suggestion and have rectified errors in the manuscript. Please refer to page 12 and 13.

“Pairwise Comparison of AUCs (Figure 3)

  1. Stepwise Logistic Regression vs LACE Index: No significant difference (Mean AUC Difference: 0.01, 95% CI -0.026-0.046, p>0.05).
  2. Stepwise Logistic Regression vs LASSO: No significant difference (Mean AUC Difference: -0.01, 95% CI: -0.066 – 0.047, p>0.05).
  3. Stepwise Logistic Regression vs Random Forest: No significant difference (Mean AUC Difference: 0.02, 95% CI: -0.010 - 0.049, p>0.05).
  4. Stepwise Logistic Regression vs XGBoost: Significant difference (Mean AUC Difference: 0.07, 95% CI 0.040 - 0.099, p<0.05).
  5. Stepwise Logistic Regression vs Neural Network: No significant difference (Mean AUC Difference: 0.02, 95% CI -0.020 – 0.060, p>0.05).
  6. LACE Index vs LASSO: No significant difference (Mean AUC Difference: 0.02, 95% CI: -0.076 – 0.036, p>0.05).

The DeLong test showed significant differences between Stepwise Logistic Regression and XGBoost (p < 0.05), while no significant differences were found between Stepwise Logistic Regression and LACE Index, Stepwise Logistic Regression and LASSO, Stepwise Logistic Regression and RF, Stepwise Logistic Regression and Neural network, or between LACE Index and LASSO (p>0.05) (Figure 3).”

Harmonise model performance reporting:

Ensure that the AUC and CI for each model in the Abstract, Results, and Discussion sections are identical.

Response: We have now corrected these errors and results of AUCs are consistent in all sections of this manuscript.

Correct the typographical error in the Discussion (RF ≠ 0.50, should be 0.60).\

Response: This error has been rectified.

Revise Table 2:

Remove duplicate sodium entries, confirm exact CIs, and include all predictors retained by the final LR model.

Clearly state the total number of variables and their selection criteria in the table legend.

Align p-values throughout. Wherever Table 1 shows p < 0.001, mirror that in the narrative.

Response: Table 2 has been revised and these errors are now rectified.

Upgrade missing-data handling: implement multiple imputation (e.g., MICE) and report the proportion of missingness per variable. Perform sensitivity analyses to assess the impact on model performance.

Response: We have now updated missing-data by providing an additional supplementary table displaying the proportion of missing data per variable. In addition, we have also performed multiple imputation using MICE, as suggested by the reviewer as a sensitivity analysis to assess the impact on model performance and have reported this in the methods and results section. Please refer to pages 3-6 under materials and methods sections and page 10 under Results section.

“The proportion of missing data for each variable is presented in Supplementary Table 1.”

“Sensitivity Analysis

We performed a sensitivity analysis by imputing missing data using the Multivariate Imputation by Chained Equations (MICE) approach via IterativeImputer from scikit-learn. A single imputed dataset was generated and used to refit all predictive models, including logistic regression, random forest, LASSO, XGBoost, and neural networks. Model performance, including AUC, was evaluated on the imputed data to assess the robustness of findings to missing data.”

Sensitivity analyses were performed using a dataset imputed via multiple imputation to assess the robustness of the model findings. Overall, the results were consistent with the primary analysis. The stepwise backward logistic regression model demonstrated an area under the receiver operating characteristic curve (AUC) of 0.63 (95% CI: 0.62–0.63), with a Hosmer–Lemeshow χ² statistic of 13.56 (p = 0.094), indicating good calibration. The LACE index yielded an AUC of 0.61 (95% CI: 0.58–0.63) and a Hosmer–Lemeshow χ² of 13.79 (p = 0.087). LASSO regression produced an AUC of 0.63 (95% CI: 0.59–0.68) with a Hosmer–Lemeshow χ² of 15.62 (p = 0.048), suggesting marginal calibration. The random forest model achieved an AUC of 0.62 (95% CI: 0.58–0.66) and excellent calibration (Hosmer–Lemeshow χ² = 2.05, p = 0.979). XGBoost resulted in a lower AUC of 0.57 (95% CI: 0.54–0.60) with a Hosmer–Lemeshow χ² of 15.5 (p = 0.050). Lastly, the artificial neural network (ANN) model demonstrated an AUC of 0.60 (95% CI: 0.58–0.62) and acceptable calibration (Hosmer–Lemeshow χ² = 9.75, p = 0.283).

Add calibration assessment:

Include calibration plots (e.g., calibration belt or LOESS) and/or Hosmer–Lemeshow statistics for each model.

Response: We have now added Hosmer-Lemeshow statistics for model calibration as per reviewer suggestion. Please refer to pages 4-6 under Methods section and pages 9-11 under Results section

Stepwise backward logistic regression

“Model performance was evaluated using the AUC and the Brier score while model calibration was assessed using the Hosmer–Lemeshow goodness-of-fit test.”

LASSO

“A fixed random seed (random_state=42) ensured reproducibility across all model training and validation procedures. Model performance was subsequently evaluated on a held-out test set (20% of the data) using the AUC and the Brier score and model calibration was assessed using the Hosmer–Lemeshow goodness-of-fit test.”

RF

“Model performance was assessed via mean AUC across 10 folds, with 95% confidence intervals and model calibration was evaluated using the Hosmer–Lemeshow test.”

XGBoost

“Model performance was assessed using AUC and Brier score, and calibration was evaluated using the Hosmer–Lemeshow goodness-of-fit test.”

ANN

“Calibration was assessed using the Hosmer–Lemeshow goodness-of-fit test.”

Results section

LACE index

“The area under the ROC curve (AUCROC) for the LACE index was 0.61 (95% CI 0.58-0.63) (Figure 2). The Brier score was 0.15 (95% 0.14-0.16) indicating reasonable predictive accuracy and model calibration was acceptable with Hosmer–Lemeshow goodness-of-fit test, χ² = 13.76, p = 0.0883.” Stepwise backward LR Model calibration was acceptable, with a Hosmer–Lemeshow goodness of fit test yielding χ² = 12.59, p = 0.1266.” 

LASSO

“AUCROC of 0.63 (95% CI 0.58-0.68) with Hosmer–Lemeshow goodness of fit test yielding χ² = 13.74, p = 0.185.”

RF

“The Random Forest model yielded a 10-fold cross-validated mean AUCROC of 0.60 (95% CI: 0.58–0.68), indicating moderate discriminative performance. However, model calibration was suboptimal, with a statistically significant Hosmer–Lemeshow test result (χ² = 17.32, p = 0.0269), suggesting discrepancies between predicted and observed event probabilities. Calibration improved modestly with Isotonic regression (χ² = 16.76, p = 0.0327), though the result remained statistically significant, indicating residual miscalibration.”

XGBoost

“XGBoost had an AUC of 0.55 (95% CI 0.53-0.56) Hosmer-Lemeshow goodness of fit test yielding χ² = 608.90, p <0.05. After recalibration using Platt scaling, the XGBoost model demonstrated good calibration, with a Hosmer–Lemeshow test result of χ² = 4.70, p = 0.7886, indicating no significant difference between predicted and observed probabilities across deciles.”

ANN

“ANN yielded an AUC of 0.60 (95% CI 0.57-0.63) with reasonable calibration Hosmer–Lemeshow goodness of fit test yielding χ² = 14.77, p = 0.0639.”

Discuss any miscalibration and potential need for recalibration before clinical use.

Response: We agree that model calibration is essential for clinical applicability. We identified miscalibration in the initial RF and XGBoost models, as indicated by statistically significant Hosmer–Lemeshow test results. Recalibration techniques were therefore applied.

Specifically, the RF model achieved a 10-fold cross-validated mean AUC of 0.60 (95% CI: 0.58–0.68), but the Hosmer–Lemeshow test indicated poor calibration (χ² = 17.32, p = 0.0269). Recalibration using isotonic regression modestly improved calibration (χ² = 16.76, p = 0.0327), although residual miscalibration persisted.

The XGBoost model had an AUC of 0.55 (95% CI: 0.53–0.56), with significant miscalibration (χ² = 608.90, p < 0.001). After applying Platt scaling, calibration improved substantially (χ² = 4.70, p = 0.7886), suggesting adequate fit.

These findings are discussed in detail in pages 10-11 under results section of the manuscript. We have now clarified the necessity of recalibration for clinical implementation.

“The Random Forest model yielded a 10-fold cross-validated mean AUCROC of 0.60 (95% CI: 0.58–0.68), indicating moderate discriminative performance. However, model calibration was suboptimal, with a statistically significant Hosmer–Lemeshow test result (χ² = 17.32, p = 0.0269), suggesting discrepancies between predicted and observed event probabilities. Calibration improved modestly with Isotonic regression (χ² = 16.76, p = 0.0327), though the result remained statistically significant, indicating residual miscalibration.

XGBoost had an AUC of 0.55 (95% CI 0.53-0.56) Hosmer-Lemeshow goodness of fit test yielding χ² = 608.90, p <0.05. After recalibration using Platt scaling, the XGBoost model demonstrated good calibration, with a Hosmer–Lemeshow test result of χ² = 4.70, p = 0.7886, indicating no significant difference between predicted and observed probabilities across deciles.”

Replace or supplement stepwise regression with a more robust variable-selection strategy. At minimum, discuss the limitations of stepwise methods and the potential for overfitting.

Response: We thank the reviewer for this valuable suggestion. We acknowledge the limitations of stepwise regression, including its reliance on arbitrary significance thresholds, potential for model instability, and risk of overfitting—particularly in datasets with correlated predictors or modest sample sizes. Stepwise methods can also lead to selection bias and underestimate the uncertainty associated with variable inclusion.

To address this, we compared the stepwise logistic regression with LASSO, a regularisation-based method known for its robustness in high-dimensional settings and its ability to prevent overfitting. The LASSO model was developed using 10-fold cross-validation to enhance generalisability and identify stable predictors. The variables selected by LASSO were largely consistent with those identified via stepwise regression, which increases our confidence in their relevance.

We have now discussed limitations of stepwise logistic regression in terms of overfitting in the limitations section of this manuscript. Please refer to page 15 under limitations section.

“Additionally, we employed stepwise logistic regression for variable selection in one of our models. While commonly used, stepwise methods have well-documented limitations, including reliance on arbitrary significance thresholds, increased risk of overfitting, and model instability—particularly in the presence of multicollinearity or modest sample sizes. These approaches may also lead to biased estimates and understate the true variability in predictor selection. To address this, we supplemented stepwise regression with LASSO regularisation, which offers a more robust and data-driven method of variable selection by penalising model complexity and reducing overfitting. The consistency of key predictors across both approaches strengthens the reliability of our findings.”

Detail hyperparameter tuning for ML models: specify search ranges, optimization criteria, and random seeds to ensure reproducibility.

Response: We have now provided detailed hyperparameter tuning for ML models, with search ranges, optimisation criteria and random seeds to ensure reproducibility as suggested by the reviewer. Please refer to pages 4-6 under methods section.

LASSO

“For this model, continuous variables were standardised using z-score normalisation. To identify the optimal regularisation strength (λ), a 10-fold stratified cross-validation was performed using grid search across a logarithmically spaced range of inverse regularisation values (C = 10⁻⁴ to 10²). The logistic regression model was fitted using the saga solver, which is appropriate for L1-penalised models. The primary optimisation criterion was the AUC, with mean AUC across folds used to select the optimal hyperparameter.

A fixed random seed (random_state=42) ensured reproducibility across all model training and validation procedures. Model performance was subsequently evaluated on a held-out test set (20% of the data) using the AUC and the Brier score and model calibration was assessed using the Hosmer–Lemeshow goodness-of-fit test.”

RF

“Hyperparameter tuning was performed using 10-fold stratified cross-validation and a grid search approach. The tuning grid included: n_estimators [100, 200, 300], max_depth [5, 10, 15], min_samples_split [2, 5, 10], and min_samples_leaf [1, 2, 4]. The random seed was fixed at 42 to ensure reproducibility. Model performance was assessed via mean AUC across 10 folds, with 95% confidence intervals and model calibration was evaluated using the Hosmer–Lemeshow test.”

XGBoost

“Hyperparameter tuning was performed using RandomizedSearchCV with 10-fold cross-validation, optimising for AUC. Key parameters tuned included n_estimators (100–1000), max_depth (3–10), learning_rate (0.01–0.3), subsample and colsample_bytree (0.5–1.0), gamma (0–5), min_child_weight (1–10), and scale_pos_weight (1–5) to address class imbalance. One hundred parameter combinations were tested using a fixed random seed (42). The best-performing model was evaluated on a held-out test set. Model performance was assessed using AUC and Brier score, and calibration was evaluated using the Hosmer–Lemeshow goodness-of-fit test.”

ANN

“Hyperparameter tuning was performed using RandomizedSearchCV with 10-fold cross-validation. Parameters included: number of hidden layers (1 or 2), units per layer (16, 32, 64, 128), dropout rate (0.2–0.5), learning rate (0.0001–0.01, log scale), batch size (16, 32, 64), and epochs (50 or 100). A fixed random_state=42 ensured reproducibility. The model used the Adam optimiser with binary cross-entropy loss and was evaluated using AUC. Calibration was assessed using the Hosmer–Lemeshow goodness-of-fit test.”

Clarify cohort derivation:

The text states an initial cohort of 5,688 with 23.8% readmissions, then excludes 317 for multiple readmissions. Spell out reasons for exclusion and assess whether excluding multiple‐readmitted patients biases the sample.

Response: We thank the reviewer for the comment. We have clarified the derivation of the final study cohort and addressed the rationale for excluding patients with multiple readmissions. Patients with multiple unplanned readmissions within 30 days (n = 317) were excluded to ensure a consistent, one-outcome-per-patient structure required for supervised machine learning and logistic regression models. Including these patients could lead to outcome dependence and inflate performance metrics due to within-subject correlation. We acknowledge this exclusion may introduce bias, as patients with frequent readmissions could differ systematically from others. This limitation has now been discussed in the manuscript. Please refer to page 7, results section and page 14 under limitations section.

“During the study period, 5,688 patients were admitted under general medicine, with 1,341 (23.8%) experiencing a readmission within 30 days of discharge. We excluded 317 patients who had multiple readmissions (>1) to ensure only a single outcome per patient, facilitating model consistency and avoiding outcome dependence. The final cohort included 5,371 patients, including 1,024 (19.1%) readmissions (Figure 1).”

“We excluded patients with multiple 30-day readmissions to maintain a single outcome per patient; however, this may introduce bias, as these patients may have distinct clinical or social characteristics influencing readmission risk.”

Expand the Limitations section to discuss single-centre design, potential selection bias, and the absence of external validation.

Response: The limitations section has been expanded as per reviewer’s suggestion. Please refer to pages 14-15.

“This study was conducted in a single tertiary care hospital, limiting generalisability to other healthcare settings, with different patient populations, care models and resource availability.”

“Our models were developed and internally validated using 10-fold cross-validation on the same dataset, but no external validation was performed. Without testing on an independent cohort, the generalisability and transportability of our models to other settings remain uncertain.”

Round 2

Reviewer 2 Report

Comments and Suggestions for Authors

Thank you for addressing the comments